# Decision Tree Analyses for Prediction of QoL over a One-Year Period in Breast Cancer Patients: An Added Value of Patient-Reported Outcomes

**DOI:** 10.3390/cancers15092474

**Published:** 2023-04-26

**Authors:** Magdalena Anna Lazarewicz, Dorota Wlodarczyk, Randi Johansen Reidunsdatter

**Affiliations:** 1Department of Health Psychology, Medical University of Warsaw, 00-581 Warsaw, Poland; magdalena.lazarewicz@wum.edu.pl (M.A.L.); dorota.wlodarczyk@wum.edu.pl (D.W.); 2Department of Circulation and Medical Imaging, Norwegian University of Science and Technology, 7030 Trondheim, Norway

**Keywords:** breast cancer, radiotherapy, patient-reported outcomes, quality of life, cancer-related symptoms, decision tree analyses

## Abstract

**Simple Summary:**

Despite the fact that self-rated health is an established and independent predictor for future health outcomes, patient-reported outcomes (PROs) are rarely utilized in clinical decisions. Such an approach can be a disadvantage for patients who might benefit from the early detection of subtle signs of deterioration in the quality of life. The present study explored the determinants of various quality-of-life trajectories during the first year after breast cancer treatment. We recognized three distinct trajectories of global quality of life: ‘high’, ‘U-shape’ and ‘low’. Our results indicate that taking PROs into account allows for a more accurate prediction of a given quality-of-life trajectory than considering only medical and sociodemographic characteristics. Concentrating on the patient’s perspective in the clinical interview is recommended, especially for patients with permanent or fluctuating lower quality of life.

**Abstract:**

Despite the current shift in medicine towards patient-centered care, clinicians rarely utilize patient-reported outcomes (PROs) in everyday practice. We examined the predictors of quality- of-life (QoL) trajectories in breast cancer (BC) patients during the first year after primary treatment. A total of 185 BC patients referred for postoperative radiotherapy (RT) filled in the EORTC QLQ-C30 Questionnaire assessing global QoL, functioning and cancer-related symptoms before starting RT; directly after RT; and 3, 6 and 12 months after RT. We used decision tree analyses to examine which baseline factors best allowed for predicting the one-year trajectory of the global QoL after BC treatment. We tested two models: ‘basic’, including medical and sociodemographic characteristics, and ‘enriched’, additionally including PROs. We recognized three distinct trajectories of global QoL: ‘high’, ‘U-shape’ and ‘low’. Of the two compared models, the ‘enriched’ model allowed for a more accurate prediction of a given QoL trajectory, with all indicators of model validation being better. In this model, baseline global QoL and functioning measures were the key discriminators of QoL trajectory. Taking PROs into account increases the accuracy of the prediction model. Collecting this information in the clinical interview is recommended, especially for patients with lower QoL.

## 1. Introduction

Breast cancer is a disease which exposes patients and survivors to long-term physical and psychological consequences. Life-saving oncological treatment brings extensive effects in the form of pain, fatigue, insomnia, and significant changes in lifestyle and social relationships, which may result in elevated levels of worry, anxiety and depression. Changes affecting the body (removal of breast, hair loss, visible scars or skin discoloration) are critical for mental representation of one’s own body, satisfaction with it and sensitivity to the opinion of others [1].

The current shift in medicine towards patient-centered care is related to the growing interest in using patient-reported outcomes (PROs), which encompass a wide range of measurable outcomes from the patient’s perspective [2]. PROs include hierarchically related perceptions of clinical symptoms, physical and psychological functional status, general health perception and global quality of life (QoL) [3]. According to the bottom-up approach to QoL, these partial elements contribute to global QoL [4]. Despite the fact that self-rated health is an established and independent predictor for future health outcomes, patient-reported outcomes (PROs) are rarely utilized for support in clinical decisions [5,6,7]. Clinicians still continue to rely on biomedical and routinely collected data. This approach can be a disadvantage for patients who might benefit from early detection of subtle signs of deterioration.

This issue is particularly relevant to breast cancer (BC) patients who, in most developed countries, receive proper healthcare and constitute the largest group of cancer survivors [8]. Thus, tertiary prevention encompassing restoration and maintenance of good QoL after treatment becomes the key priority [9,10,11,12,13]. This positive trend was confirmed in longitudinal studies estimating QoL on the basis of average-level data [9,10,11,13,14,15] or in subgroups defined by treatment modalities [16,17]. However, in the studies investigating more individualized clusters of QoL trajectories, not only patients with high- or medium-improving QoL were recognized but also smaller but significant groups of patients with low-deteriorating QoL over the period of 1 year after treatment [12], ‘accelerated decline’ within 7 years [18] or decline after 1 year within a 3-year observation [19]. There are still only a few prospective studies evaluating the QoL of patients during and after radiotherapy (RT), and they are noticeably outnumbered by those assessing patients receiving chemotherapy [16].

The QoL of BC patients is strongly influenced by factors such as stage and complexity of the disease and effects and side effects of treatment [20,21], but little is known about the configuration of factors contributing to QoL trajectories over time. In our research, we aimed to verify the goodness of prediction for two models. The first model (basic) included only biomedical data, patient characteristics routinely collected during the medical interview and baseline global QoL (for control). The second model (enriched) additionally included partial aspects of PROs in the form of self-reported functioning and symptoms specifically important in BC patients. We wanted to estimate the value of adding partial aspects of PROs to allow for the best prediction of the trajectory of the QoL of BC patients.

## 2. Patients and Methods

### 2.1. Study Group and Treatment

BC patients referred for postoperative RT were consecutively informed about and invited to a follow-up study investigating QoL after RT. The exclusion criteria were metastatic disease, physical or psychological disorders that would interfere with participation or being unable to speak and understand Norwegian. Out of 261 eligible patients, 250 (96%) patients consented to participate.

Standardized treatment was provided in accordance with Norwegian national guidelines in force during the data collection (www.nbcg.no/arkivlenker (archived guidelines IS-1524 accessed on 15 April 2023)). Chemotherapy was administered routinely prior to RT as anthracycline-based courses, followed by docetaxel or paclitaxel. Conventional CT-planned RT was delivered to the breast/chest wall in 2 Gy/fractions, 5 days a week for a total dose of 50 Gy. According to hormonal receptor status, adjuvant endocrine therapy was scheduled for 5–10 years.

### 2.2. Study Procedure

Patients were recruited for the study in the period between February 2007 and October 2008 and followed for one year through five extended outpatient controls at the hospital: before starting RT (baseline), directly after RT (T1), and 3, 6 and 12 months after RT (T2–T4) [12,22]. At each control, patients were examined by an oncologist, underwent several diagnostic examinations and delivered standardized PROMs. For the present paper, only the PROMs were used in conjunction with the clinical baseline data.

### 2.3. Measures

At baseline, sociodemographic data were collected from patients by self-report questionnaires, and treatment and clinical variables were registered by the oncologist (see Table 1 for details).

At all assessments, global QoL and partial PROs (functioning and symptoms) were measured by subscales of the European Organization for Research and Treatment of Cancer Quality of Life Questionnaire (EORTC QLQ-C30) [23]. The *Global QoL* subscale includes two items on overall health and QoL during the past week with response options ranging from 1 (very poor) to 7 (excellent). *Functional subscales* cover physical (five items), cognitive (two items), emotional (four items), role (two items) and social (two items) functioning during the previous week. The *symptom subscales* comprise pain (two items covering the presence of pain and its interference with daily activities), fatigue (three items addressing tiredness, weakness and lack of energy) and insomnia (one item) during the previous week. Response options for functional and symptom subscales ranged from 1 (not at all) to 4 (very much). All subscales were calculated according to the EORTC scoring manual as the average score and transformed to a 0–100 scale, with higher scores indicating better global QoL and functioning, as well as more symptoms [24]. Cronbach’s alpha values for the utilized EORTC QLQ-C30 scales were as follows: global QoL 0.90, physical, role, emotional, cognitive and social functioning 0.77, 0.90, 0.84, 0.65 and 0.78, and for the symptom scales, fatigue and pain 0.90 and 0.86, respectively.

### 2.4. Statistical Analyses

We used hierarchical cluster analysis to distinguish subgroups with different trajectories of QoL over the first year after RT (T1 to T4). In these analyses, only patients with responses at each assessment were utilized (n = 185). Cluster analysis allows for classifying cases that are relatively homogeneous within themselves and relatively heterogeneous between each other [25]. We used a squared Euclidean distance as a distance measure and complete linkage as a linkage method. The decision regarding the final number of clusters of trajectories was made based on the agglomeration schedule and the dendrogram; we chose the three-cluster solution [12]. We checked the differences between clusters using one-way ANOVA, the Kruskal–Wallis test with Mann–Whitney U test as a post hoc test, the chi-square test and Cramer’s V test, according to the level of variables [26].

To analyze the nature of missingness, we used Little’s MCAR test, which confirmed that missingness was completely at random (*p* > 0.05). Thus, we assumed that the obtained results are representative of the initial sample.

To determine which baseline factors allowed for prediction of the probability of the occurrence of one of the three trajectories of the QoL of BC patients over the first year after RT (T1 to T4), we used decision trees (the tool for classification and prediction based on decision rules learned and inferred from the data features) [27]. Due to the relatively small sample size, we used the method for exploratory purposes. It has a flowchart-like structure, including nodes (representing a test on an attribute), branches (representing the outcome of the test) and leaves, with the values of the target variable. An instance is classified by starting at the root node of the tree, testing the attribute specified by this node and then moving down the tree branch corresponding to the value of the attribute. The predictor with the highest association with the target variable was selected for splitting. The process was repeated recursively until predefined stopping rules are triggered. To evaluate the quality of the classification, we used the index of risk and indexes of accuracy, sensitivity and specificity calculated based on the classification matrix. We also used these indexes for the qualitative comparison of the two tested models.

## 3. Results

### 3.1. Baseline Characteristics of the Sample

The baseline sociodemographic and medical characteristics and the level of functioning, symptoms and global QoL in the initial sample are presented in Table 1.

**Table 1 cancers-15-02474-t001:** Baseline demographic and medical characteristics, function, symptoms and global QoL of the sample (N = 185).

Demographic Characteristics	N (%)	Medical Characteristics	N (%)
Mean age (SD, range)	57.4 (9.0, 28–79)	AJCC stage	
Marital status		Stage 0	15(8.1)
Living alone	41 (22.2)	Stage I	95 (51.4)
Married/cohabiting	143 (77.3)	Stage IIA	43 (23.2)
Missing	1 (0.5)	Stage IIB	16 (8.6)
Education		Stage III	16 (8.6)
Primary school (7–10 grade)	48 (25.9)	Comorbidity (Yes)	48 (25.9)
Vocational (1–2 grades)	59 (31.9)	Cardiovascular	26 (14.0)
High school (2–4 grades)	24 (13.0)	Pulmonary	7 (3.8)
University, 3 years	23 (12.4)	Other	15 (8.1)
University >3 years	29 (15.7)	Breast surgery	
Missing	2 (1.1)	Conservative	134 (72.4)
Employment		Radical	51 (27.6)
No *	129 (69.7)	Radiotherapy	
Yes	52 (28.1)	Local	122 (65.9)
Missing	4 (2.2)	Locoregional	63 (34.1)
Annual family income in NOK		Chemotherapy (Yes)	77 (41.6)
<300,000	31 (16.8)	Endocrine therapy (Yes)	102 (55.1)
300,000–499,000	53 (28.6)	Trastuzumab (Yes)	24 (13.0)
>500,000	84 (45.4)		
Missing	17 (9.2)		
**Function**	**Mean (SD, range)**	**Symptoms**	**Mean (SD, range)**
Physical Function	87.0 (15.7, 26.7–100)	Fatigue	25.6 (23.4, 0–100)
Role Function	79.1 (26.8, 0–100)	Pain	14.9 (23.2, 0–100)
Emotional Function	81.6 (18.9, 25–100)	Insomnia	24.6 (28.6, 0–100)
Cognitive Function	86.4 (18.4, 0–100)		
Social Function	79.0 (23.6, 0–100)	Global QoL	75.7 (19.8, 16.7–100)

Note. SD—standard deviation; NOK—Norwegian krone; AJCC stage—stage of cancer according to the American Joint Committee on Cancer classification system. * This category includes unemployed participants as well as those on sick leave and retired. The European norm score for females for the EORTC QLQ-C30 scales are, (M (SD): physical function 89.4 (15.0), role function 85.0 (24.3), emotional function 78.2 (20.9), cognitive function 81.6 (20.1), social function 83.3 (24.2), fatigue 33.9 (24.8), pain 24.1 (26.6), insomnia 30.2 (29.8) and global QoL 71.1 (22.3) [28].

### 3.2. Clusters of Global QoL Trajectories

The three selected clusters of trajectories of global QoL can be described as ‘high’ (n = 130, 70%), ‘U-shape’ (n = 27, 15%) and ‘low’ (n = 28, 15%; Figure 1 and Table 2). The ‘high’ cluster showed a rather stable QoL over the T1-T4 period, with mean scores (SD) ranging from 70 (18) to 87 (12) and with a small but significant increase from T1 to T2, T3 and T4. The ‘U-shape’ cluster can be described as U-falling, with QoL scores significantly lower at T2 and T3 than at T1 and T4; mean scores (SD) ranged from 60 (17) to 77 (15). The ‘low’ cluster showed a stable QoL over the T1-T4 period, with mean scores (SD) ranging from 45 (16) to 50 (14). The three clusters of global QoL differed significantly at all measurements, except for at T1, where there were no significant differences between the ‘high’ and ‘U-shape’ clusters (t = 1.94 (57.4), *p* = 0.057).

### 3.3. Baseline Characteristics and Differences between Clusters of Global QoL Trajectories

Baseline characteristics of the clusters of global QoL trajectories and differences between them are presented in Table 3 and Table 4. There were hardly any significant differences between patients in the clusters in terms of most ‘objective’ factors, that is, in sociodemographic and medical characteristics. The only difference was in the proportion of patients undergoing conservative vs. radical surgery (Cramer’s V = 0.192, *p* = 0.033), with a higher number of patients receiving conservative surgery in both the ‘high cluster’ and ‘low cluster’ when compared to that in the ‘U-shape cluster’.

Considering partial PROs (functioning and symptoms) and global QoL at baseline (Table 4), the patients in the ‘high’ and the ‘U-shape’ clusters scored significantly better (in terms of higher functioning, lower symptoms and higher global QoL) than patients in the ‘low’ cluster. Role functioning was an exception, with patients in the ‘high’ cluster scoring higher than patients in both the ‘U-shape’ and ‘low’ clusters. The statistically significant differences between clusters in global QoL, functioning and symptoms were also clinically significant, with differences between mean points ranging from 13 to 29 points (according to [29], a mean score difference of 5–10 is usually regarded as a small but clinically noticeable change, a change between 10 and 20 as moderate and >20 as a large clinical difference).

### 3.4. ‘Basic’ Model—Decision Tree Analysis

The analysis of the ‘basic’ decision tree, including 13 candidate predictors (sociodemographic variables (age, marital status, education, income and employment), clinical variables (stage of cancer according to the American Joint Committee on Cancer (AJCC stage)), comorbidities, surgery, chemotherapy, extent of RT, hormonal therapy and trastuzumab) and global QoL at baseline showed that the probability of occurrence of a given QoL trajectory is significantly affected by *global QoL at baseline*, *AJCC stage*, *income* and *age* (Figure 2).

The baseline global QoL was the strongest discriminator between the three trajectories. In patients with a baseline QoL ≤ 29.2 (the left split, the second node), the probability of a ‘low’ QoL trajectory was 2.9 times higher than in patients with a baseline QoL between 29.2 and 70.8 (83.3% vs. 28.8%) and 23.8 times higher than in patients with a baseline QoL > 70.8 (3.5%). However, for patients with a baseline QoL between 29.2 and 70.8, lower income (≤300,000 NOK) significantly increased the probability of a ‘low’ (50.0% vs. 23.1%) and ‘U-shape’ (21.4% vs.13.5%) QoL trajectory.

Among patients with a baseline QoL > 70.8 (the right split), the probability of a given trajectory was significantly influenced by AJCC stage and age. In patients with a more advanced stage of disease (IIb and III), the probability of a ‘high’ QoL trajectory was 1.6 times lower than in patients with a less advanced stage of disease (AJCC stage 0, I, IIa; 53.8% vs. 86%), and the probabilities of ‘U-shape’ and ‘low’ QoL trajectories were, respectively, 1.8 (23.1% vs. 13.0%) and 23.1 (23.1% vs. 1.0%) times higher. Among patients at a less advanced stage of the disease (AJCC stage 0, I, IIa), the probability of a given trajectory was additionally influenced by age, with the probability of a ‘U-shape’ QoL trajectory being 4.4 times higher in patients over 52.5 years old (16.4%) when compared to that in patients ≤ 52.5 years old (3.7%).

### 3.5. ‘Enriched’ Model—Decision Tree Analysis

The analysis of the ‘enriched’ decision tree included 21 candidate predictors, including all variables from ‘basic’ decision tree) and baseline self-reported functioning (social, emotional, physical, cognitive and role) and symptoms (pain, fatigue and insomnia). Figure 3 shows that the probability of the occurrence of a given QoL trajectory is significantly affected by *social, emotional and cognitive functioning*, *global QoL at baseline, income* and *AJCC stage*.

The strongest discriminator between the three trajectories was baseline social functioning. In patients with social functioning ≤58.3 (the left split), the most probable was the ‘low’ QoL trajectory (44.4%), while in patients with social functioning >58.33, the most probable was the ‘high’ QoL trajectory (77.2%). Among patients with lower social functioning (≤58.3), the emotional functioning ≤70.8 significantly increased the probability of a ‘low’ QoL trajectory (85.7% vs. 0%). The probability of ‘high’ and ‘U-shape’ QoL trajectories was almost the same (53.8% vs. 46.2%) in patients with emotional functioning >70.8; however, the coexistence of higher emotional functioning (>70.8) and lower cognitive functioning (≤75.0) increased the probability of a ‘U-shape’ QoL trajectory (80% vs. 25%) and decreased the probability of a ‘high’ QoL trajectory (20% vs. 75%).

Among patients with social functioning >58.3 (the right split), the probability of a given trajectory was significantly influenced by baseline QoL, income and AJCC stage. Among those with a baseline QoL ≤ 70.8, the probability of a ‘high’ QoL trajectory was 2.7 times lower in patients with an income below 300,000 NOK (27.3%) than in patients with an income above this amount (73.2%). The probability of ‘U-shape’ and ‘low’ trajectories were respectively 2.8 (27.3% vs. 9.8%) and 2.7 (45.5% vs. 17.1%) times higher in patients with lower income. In patients with a baseline QoL > 70.8, the probability of a ‘high’ QoL trajectory was 1.8 times lower in patients in stage IIB or III (50%) than in stage 0, I or IIA (88.3%), and the probability of a ‘low’ QoL trajectory was 23 times higher (25% vs. 1.1%). More advanced stages of disease also increased the probability of a ‘U-shape’ QoL trajectory (25% vs. 10.6%).

### 3.6. Comparison of the ‘Basic’ and ‘Enriched’ Decision Trees

To compare the quality of the decision trees, we performed a qualitative analysis based on the indexes of quality. The index of risk, indicating the probability of wrong classification, was 0.25 (SE = 0.03) for the ‘basic’ tree and 0.21 (SE = 0.03) for ‘enriched’ tree. Compared to the ‘basic’ tree, the ‘enriched’ tree improved the classification accuracy from 77% to 81% for the ‘high’ QoL trajectory, from 15% to 87% for the ‘U-shape’ QoL trajectory and from 15% to 90% for the ‘low’ QoL trajectory. Regarding sensitivity (percentage of correctly classified positive cases), there was no difference for the ‘high’ QoL trajectory (96%), but for the ‘U-shape’ QoL trajectory, it changed from 0% to 15% and for the ‘low’ QoL trajectory, from 43% to 60%. Regarding specificity (percentage of correctly classified negative cases), there were no differences for the ‘U-shape’ and ‘low’ QoL trajectories (99% and 95%, respectively), but for the ‘high’ QoL trajectory, it improved from 29% to 45%.

## 4. Discussion

This study analyzed two different approaches to predict the QoL trajectories of BC patients during the first year after postoperative RT. Drawing on results pointing to the heterogeneity of QoL changes over time [12], we distinguished three QoL trajectories and found that 30% of patients had temporarily or permanently low QoL. This finding was in line with other studies conducted in patients after BC diagnosis, showing higher diversity in QoL variation than in the studies examining only average changes [18,19]. Similar to these studies, in our sample, baseline sociodemographic and clinical characteristics did not differ between clusters, with the exception of some aspects of treatment. This may suggest that, to understand QoL trajectories, other explaining variables (such as PROs) should be considered.

Including PROs in the model improved the accuracy of classification for all trajectories. Both analyzed models were similar in the sensitivity indexes for the ‘high’ QoL trajectory, but the ‘enriched’ tree allowed for noticeably better prediction of the ‘U-shape’ and ‘low’ QoL trajectories. These are the groups that require special attention and care because they are especially prone to further deterioration. It was proved that lower baseline QoL was predictive of lower future QoL, at least from the 5-year perspective [30]. The consequences of some aspects of QoL are even more distant and extensive; for example, decline in self-reported physical functioning after BC diagnosis predicted lower 10-year survival in older women [31].

The results showed that, regardless of the predictors included in the models, there were some similarities. In both models, baseline QoL, stage of disease and income play a role. In another study of patients undergoing RT, the ability to pay for basics differentiated trajectories of physical wellbeing and cancer-specific symptoms, but stage of disease was not a conclusive determining factor [32]. The ‘enriched’ model showed that the influence of the stage of cancer manifested only in patients with better social functioning and higher baseline QoL, increasing the chance of developing a ‘high’ QoL trajectory among those with earlier stages of cancer. Simultaneously, the role of income manifested in patients with better social functioning but lower baseline QoL, increasing the chance of developing a ‘high’ QoL trajectory among those with higher income.

In the ‘enriched’ decision tree, age was not a significant discriminator between QoL trajectories, but social, emotional and cognitive functioning revealed their potential for prediction. There was also a difference between the models in the primary discriminator. Social functioning turned out to be the most important in the classification of patients to QoL trajectories. Having strong social ties and sharing experiences with another person makes people healthier and happier [33,34]. Social support resulting from stable family and social relationships may be especially important in the context of RT. As side effects of RT are not as visible as after surgery or chemotherapy, family members may not be aware of the ongoing importance of social support, while its lack or decrease seems to have a critical impact on global QoL. The ‘enriched’ model also revealed the role of cognitive functioning and showed that, in patients with lower social functioning but better emotional functioning, those reporting more problems with concentration and memory are at higher risk of developing a ‘U-shape’ QoL trajectory. Cancer-related cognitive impairment (CRCI) and its relation to QoL have been described mostly in BC patients after chemotherapy and labelled *chemobrain* [35].

Importantly, the models have the same complexity and the ‘enriched’ model did not lose its clarity, despite the greater number of predictors. Integration of PROs into clinical care seems to be a critical step to achieve optimal outcomes of treatment and healthcare. It may be achieved through better patient–provider communication, as PROs have been shown to facilitate shared decision-making [6]. Due to the work overload of doctors and nurses, lack of time for a full interview and analyzing all factors potentially related to treatment outcomes, it is essential to recognize those having the most predictive value for the QoL of patients.

The present study has some limitations. As our original study was designed to assess QoL after radiotherapy, the baseline QoL of the patients receiving chemotherapy was probably influenced by chemotherapy at the first assessment. However, this possible effect would be watered down during the following assessments. Further, due to the relatively small sample size, we used decision tree analysis for exploratory purposes without setting strict parameters to allow for verification of specific hypotheses. A bigger sample size would increase the chance of proper classification of cases and secure a satisfactory number of cases in nods. Although the ‘enriched’ model made it possible to decrease the index of risk in comparison to the ‘basic’ model, further research is needed to improve the correctness of classification. A larger sample would allow inclusion of other factors into the ‘enriched’ model that could possibly contribute to QoL trajectories, such as social support, coping mechanisms and personality dispositions, such as optimism [18] or resilience [36] or other aspects of QoL. Further, as breast cancer treatment and PROM responses [28] have not changed dramatically over years, associations included in our models and results are regarded as rather time-independent. However, it is crucial to focus on factors that might be easily collected by health care providers during medical appointments and monitored over time. Nowadays, collecting and utilizing relevant PROs in clinical practice seems to be difficult, but it may be an inevitable direction for change in future healthcare systems.

## 5. Conclusions

Both the ‘basic’ and ‘enriched’ models seem informative and helpful in recognizing BC patients with higher chances of developing specific QoL trajectories; however, the ‘enriched’ model allows for better prediction, especially for the ‘U-shape’ and ‘low’ QoL trajectories. In this more precise ‘enriched’ model, PROs are the key discriminators of global QoL trajectory. Further research in a bigger sample size with the possibility of setting strict parameters of classification is needed to confirm the above results. In the rushed everyday clinical practice, the ‘basic’ model may be applied but with an awareness of the higher risk of error in prediction of the QoL trajectory. In more favorable clinical conditions, it is recommended to include PROs in the clinical interview or via standardized tests conducted by the multidisciplinary team to recognize BC patients at risk of temporarily or permanently deteriorated QoL. Thus, efforts aimed at the reorientation of the health system into patient-centered care would be recommended. This study provokes also more general reflection regarding breast cancer patients’ active participation in clinical studies, not only to keep their motivation to take part but also at the stage of determining the scope of the research, especially if it refers to PROs [37].

## Figures and Tables

**Figure 1 cancers-15-02474-f001:**
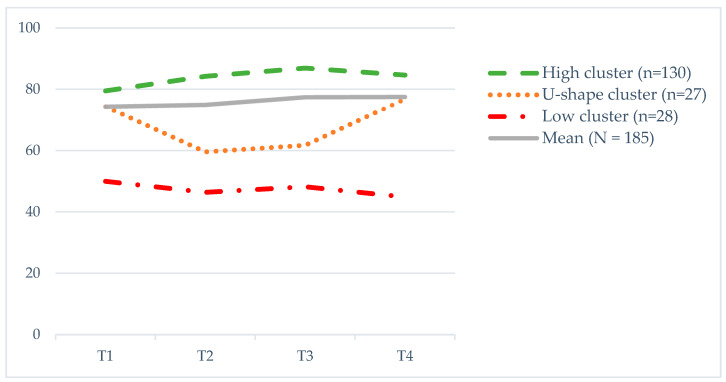
Clusters of trajectories of global QoL during one year after ending primary treatment. Measurements: T1—at the completion of RT; T2—3 months after RT; T3—6 months after RT; T4—12 months after RT. QoL—quality of life; RT—radiotherapy. The European norm score for females for the global QoL scale in EORTC QLQ-C30 is 71.1 [28].

**Figure 2 cancers-15-02474-f002:**
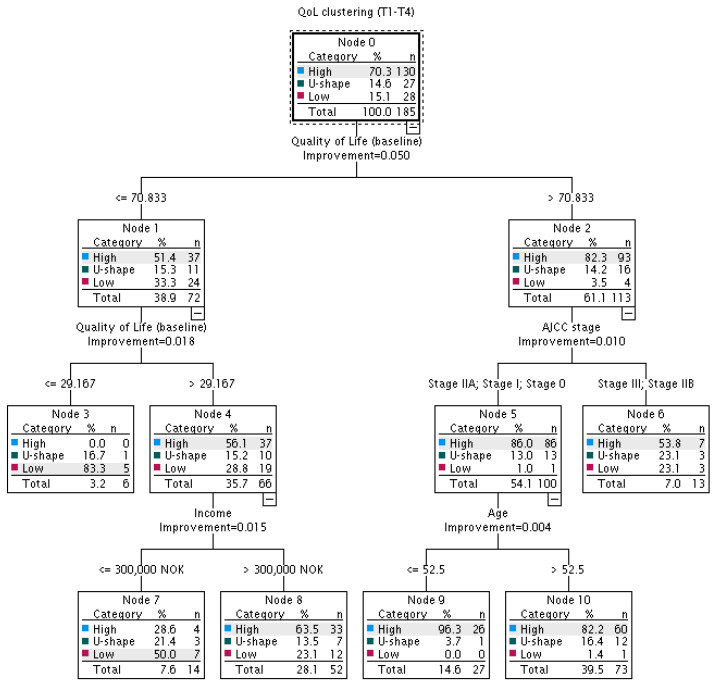
Probability of occurrence of a ‘high’, ‘U-shape’ or ‘low’ trajectory of QoL over a one-year period after radiotherapy: decision tree analysis including objective sociodemographic and clinical factors. QoL—quality of life; AJCC stage—stage of cancer according to the American Joint Committee on Cancer classification system; NOK—Norwegian krone.

**Figure 3 cancers-15-02474-f003:**
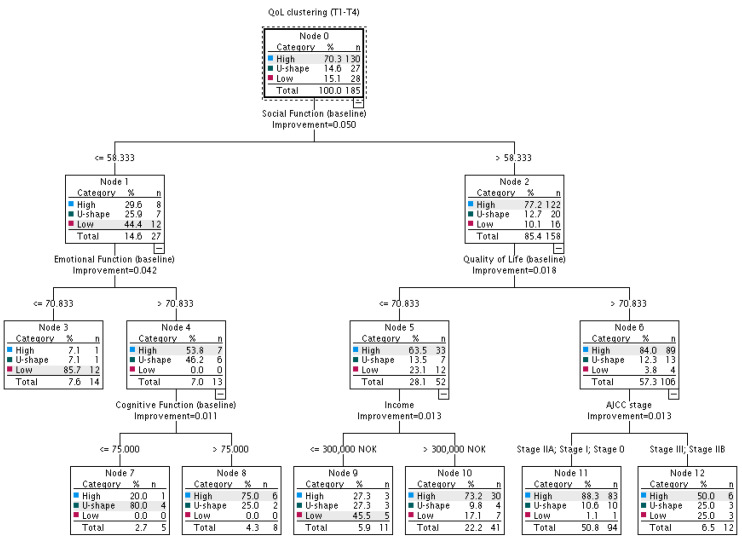
Probability of occurrence of a ‘high’, ‘U-shape’ or ‘low’ trajectory of health-related QoL over a one-year period after radiotherapy: decision tree analysis including objective sociodemographic and clinical factors and subjective QoL, function and symptoms. QoL—quality of life; NOK—Norwegian krone; AJCC stage—stage of cancer according to the American Joint Committee on Cancer classification system.

**Table 2 cancers-15-02474-t002:** Descriptive statistics and repeated-measures ANOVA with Bonferroni correction for post hoc analysis of global QoL clusters during the first year after breast cancer treatment.

	T1	T2	T3	T4	F(*p*)	Differences *
QoL cluster	M (SD)	M (SD)	M (SD)	M (SD)		
High (n = 130)	79.5 (17.6)	84.2 (13.3)	86.9 (11.5)	84.6 (14.8)	11.0 (<0.001) ^a^, eta = 0.079	T1 < T2,T3,T4
U-shape (n = 27)	74.4 (11.1)	59.6 (16.6)	61.7 (12.9)	76.9 (14.7)	10.8 (<0.001), eta = 0.293	T1,T4 > T2,T3
Low (n = 28)	50.0 (14.2)	46.4 (13.1)	48.2 (13.9)	44.9 (15.8)	0.797 (0.499)	*no sig. dif.*

QoL—quality of life; M—mean; T1—assessment at the completion of RT; T2—assessment 3 months after RT; T3—assessment 6 months after RT; T4—assessment 12 months after RT; RT—radiotherapy. ^a^ Repeated-measures ANOVA with Huynh–Feldt correction. * The mean difference is significant at the <0.05 level.

**Table 3 cancers-15-02474-t003:** Demographic and medical characteristics of breast cancer patients in high, U-shape and low QoL clusters before starting RT (N = 185).

	N(%)		
	High Cluster (n = 130)	U-Shape Cluster (n = 27)	Low Cluster (n = 28)
**Demographic Characteristics**			
Mean age (SD, range)	57.2 (8.99, 28–89)	55.2 (8.81, 36–71)	60.3 (8.84, 41–79)
Marital status			
Living alone	25 (19.2)	9 (33.3)	7 (25.0)
Married/cohabiting	104 (80.0)	18 (66.7)	21 (75.0)
Missing	1 (0.8)		
Education			
Primary school (7–10 grade)	31 (23.8)	8 (29.6)	9 (32.1)
Vocational (1–2 grades)	46 (35.4)	5 (18.5)	8 (28.6)
High school (2–4 grades)	15 (11.5)	4 (14.8)	5 (17.9)
University	36 (27.7)	10 (37.0)	6 (21.4)
Missing	2 (1.5)		
Employment			
No *	86 (66.2)	19 (70.4)	24 (85.7)
Yes	42 (2.3)	6 (22.2)	4 (14.3)
Missing	2 (1.5)	2 (7.4)	
Annual family income in NOK			
<300,000	16 (12.3)	5 (18.5)	10 (35.7)
300,000–499,000	40 (30.8)	6 (22.2)	7 (25.0)
>500,000	65 (50.0)	8 (29.6)	11 (39.3)
Missing	9 (6.9)	8 (29.6)	
**Medical characteristics**			
AJCC stage			
Stage 0	13 (10.0)	—	2 (7.1)
Stage I	67 (51.5)	14 (51.9)	14 (50.0)
Stage IIA	32 (24.6)	6 (22.2)	5 (17.9)
Stage IIB	9 (6.9)	4 (14.8)	3 (10.7)
Stage III	9 (6.9)	3 (11.1)	4 (14.3)
Comorbidity (Yes)	29 (22.3)	7 (25.9)	12 (42.9)
Breast surgery			
Conservative	98 (75.4)	14 (51.9)	22 (78.6)
Radical	32 (24.6)	13 (48.1)	6 (21.4)
Radiotherapy			
Local	89 (68.5)	13 (48.1)	20 (71.4)
Locoregional	41 (31.5)	14 (51.9)	8 (28.6)
Chemotherapy (Yes)	55 (42.3)	15 (55.6)	7 (25.0)
Endocrine therapy (Yes)	65 (50.0)	18 (66.7)	19 (67.9)
Trastuzumab (Yes)	22 (16.9)	2 (7.4)	—

QoL—quality of life; RT—radiotherapy; SD—standard deviation; NOK—Norwegian krone; AJCC stage—stage of cancer according to the American Joint Committee on Cancer classification system. * This category includes unemployed participants as well as those on sick leave and retired.

**Table 4 cancers-15-02474-t004:** Global QoL, function and symptoms in breast cancer patients in high, U-shape and low QoL clusters before starting RT (N = 185).

	Mean (SD)			Differences *
	High (n = 130)	U-Shape (n = 27)	Low (n = 28)	
Global QoL	80.8 (16.0)	72.6 (20.6)	53.7 (20.3)	H&U > L
Physical Function	89.9 (13.0)	87.5 (12.4)	72.8 (21.6)	H&U > L
Role Function	83.7 (24.0)	70.0 (28.1)	66.0 (32.5)	H > U&L
Emotional Function	84.9 (16.2)	85.4 (14.8)	62.3 (22.9)	H&U > L
Cognitive Function	89.3 (15.9)	86.1 (16.1)	72.8 (25.0)	H&U > L
Social Function	84.4 (17.6)	75.0 (24.6)	56.8 (32.8)	H&U > L
Fatigue	20.5 (20.6)	26.2 (23.3)	49.4 (22.5)	H&U < L
Pain	10.7 (17.5)	14.0 (21.3)	35.8 (25.4)	H&U < L
Insomnia	22.3 (27.6)	18.7 (21.7)	40.7 (33.8)	H&U < L

Note. * One-way ANOVA test + post hoc (with appropriate corrections if needed). QoL—quality of life; RT—radiotherapy; SD—standard deviation; H—‘high’ cluster; U—‘U-shape’ cluster; L—‘low’ cluster.

## Data Availability

The datasets generated and analyzed during the current study are not publicly available but are available from the corresponding author on reasonable request.

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
