# Peer review of "Decision Tree Analyses for Prediction of QoL over a One-Year Period in Breast Cancer Patients: An Added Value of Patient-Reported Outcomes"

_cancers, 2023, doi:10.3390/cancers15092474_

Round 1

Reviewer 1 Report

Dear authors, thank you for your interesting contribution. Here are many suggestions:

1. Why do you choose only patients after radiotherapy? Is there a scientific criterium? Please, explain in

2. Please, add more details about breast cancer and its related consequence. To date, the background lacks information about emotions, cognition, and body-relationship after cancer. Please, refer to Sebri et al., 2022; McGannon et al., 2016, for instance

3. How did you conduct the patients' recruitment? Please, be clearer about it

4. Make a specific paragraph for the questionnaires. They have to be more clarified (add alpha of Chronbach, please)

5. In the statistical analysis, you mentioned the small sample size. Please, move on to this aspect as a limitation and explain it

6. Please, add practical implications and limitations in your conclusions. For example, it is needed to consider patients' motivation to participate in experimental research as a consideration for future research. Please, mention Savioni et al., 2022

REFERENCES:

McGannon, K.R.; Berry, T.R.; Rodgers, W.M.; Spence, J.C. Breast cancer representations in Canadian news media: A critical discourse analysis of meanings and the implications for identity. Qual. Res. Psychol. 201613, 188–207

Savioni, L., Triberti, S., Durosini, I., Sebri, V., & Pravettoni, G. (2022). Cancer patients’ participation and commitment to psychological interventions: A scoping review. Psychology & Health37(8), 1022-1055.

Sebri, V., Durosini, I., Mazzoni, D., & Pravettoni, G. (2022). The Body after Cancer: A Qualitative Study on Breast Cancer Survivors’ Body Representation. International Journal of Environmental Research and Public Health19(19), 12515.

Reviewer 2 Report

The aim of the study –  to examine “the predictors of quality of life (QoL) trajectories in breast cancer (BC) patients during the first year” of postoperative radiotherapy -  is highly relevant to patients and clinicians, health systems, and researchers and policy-makers. The methodology is limited to the small sample size which only allowed for an exploratory analysis but the statistical approach is quite innovative and results are easy to interpret.

Symptoms assessed with the EORTC QLQ-C30 questionnaire were not predictors of QoL trajectories nor physical functioning, suggesting that they are well managed during primary treatment. However, other components of functioning, namely role, emotional and social functioning were predictors of QoL trajectories. This result highlights the need for assessing not only low QoL before radiotherapy but also the functioning components of QoL.

I do not think that we should advocate for the use of the QLQ-C30 (or another similar questionnaire) during nurse or physician consultations because this questionnaire could be filled in at different moments, but rather we should recommend the existence of a multidisciplinary team including psychologists and social services who could respond to the complaints reported by patients.

If the authors agree with this point of view, they could discuss this organizational structure of health services centered on the patient as a whole, and not a subject with a disease that need to be controlled.

Besides this general overview of the manuscript, I would like to point out some aspects that need to be improved:

1.       The text “The first model (basic) included only biomedical data and patient characteristics routinely collected during the medical interview. The second model (enriched) additionally included partial aspects of PROs in the form of self-reported functioning and symptoms specifically important in BC patients.” in lines 68-72 do not refer that global QoL was also included in the models.

2.       In the section “Patients and Methods”, please provide when and where was the study conducted.

3.       Please, review the references. I did not check all the references but at least in this sentence “The recruitment procedure and patient dropout are described in detail elsewhere [10, 20].”, references are not correct, as ref.10 is “10. Koch, L.; Jansen, L.; Herrmann, A.; Stegmaier, C.; Holleczek, B.; Singer, S.; et al. Quality of life in long-term breast cancer sur- vivors – a 10-year longitudinal population-based study. Acta Oncol. 2013; https://doi.org/10.3109/0284186X.2013.774461” and 20 is “20. Ou, H.T.; Chung, W.P.; Su, P.F.; Lin, T.H.; Lin, J.Y.; Wen, Y.C.; et al. Health-related quality of life associated with different cancer treatments in Chinese breast cancer survivors in Taiwan. Eur J Cancer Care. 2019; https://doi.org/10.1111/ecc.13069”

4.       Please clarify if unemployed includes sick leaves and retirement.

5.       Please, clarify “Total family income in NOK”: annual family income per capita?

6.       Why only pain, insomnia and fatigue were considered? What about the other symptoms assessed with the QLQ-C30 questionnaire?

7.       Please, correct “AJCC” to “AJCC stage” or “BC stage”. AJCC stands for American Joint Committee on Cancer.

8.       Comorbidities are important when assessing QoL and I would expect a better use of comorbidity data than simply a “yes or no” to the existence of at least one comorbidity.

9.       Why axillary lymph node dissection vs. sentinel lymph node biopsy was not considered in this analysis?

10.   In Table 4, comparisons were made using One-Way ANOVA test. Wouldn't the Kruskal-Wallis test be a better test with these variables?

11.   In Figures 2 and 3, correct QL to QoL and AJCC to AJCC stage or BC stage.

12.   In the Discussion section, lines 288-289 “…30% of patients had QoL levels indicating temporary or permanent burden of disease or treatment over this period.”: the low trajectory may be not due to cancer or its treatment as these patients could already have a low QoL even before cancer diagnosis.

13.   Please state limitations of the study other than the sample size, namely regarding the generalizability of the results, and the non-inclusion of data from the complementary questionnaire BR23 module.

14.   Please correct the sentence “To conclude, both the ‘basic’ and ‘enriched’ models seem informative and helpful in recognizing QoL trajectories in BC patients;” in lines 350-351: decision tree models were used to identify predictors of QoL trajectories and not QoL trajectories.

Round 2

Reviewer 1 Report

No other comments needed

Reviewer 2 Report

The authors have satisfactorily improved the manuscript regarding minor and major aspects, making it more precise and clear. 

I have no more comments to add.